

# ECG is an inefficient screening-tool for left ventricular hypertrophy in normotensive African children population

Giuseppe Di Gioia[1], Antonio Creta[1], Cosimo Marco Campanale[1],
Mario Fittipaldi[2], Riccardo Giorgino[1], Fabio Quintarelli[3], Umberto Satriano[1],
Alessandro Cruciani[1], Vincenzo Antinolfi[4], Stefano Di Berardino[5],
Davide Costanzo[1], Ranieri Bettini[6], Giuseppe Mangiameli[5], Marco Caricato[5] and
Giovanni Mottini[7]

[1] Department of Medicine and Surgery, Unit of Cardiology, Campus Bio-Medico University of Rome, Rome, Italy
[2] Paediatric Cardiothoracic Surgery, Starship Greenlane Paediatric and Congenital Heart Service, Auckland, New Zealand
[3] Department of Medicine and Surgery, Service of Pediatrics, Campus Bio-Medico University of Rome, Rome, Italy
[4] Heart and Great Vessels "Attilio Reale", University of Roma "La Sapienza", Rome, Italy
[5] Department of Medicine and Surgery, Geriatric Surgery Unit, Campus Bio-Medico University of Rome, Rome, Italy
[6] Cardiology Department, University of Pisa, Pisa, Italy
[7] Institute of Philosophy of Scientific and Technological Practise (FAST), Campus Bio-Medico University of Rome, Rome, Italy

Corresponding author
Giuseppe Di Gioia, dottgiuseppedigioia@gmail.com

## ABSTRACT

**Background.** Left ventricular hypertrophy (LVH) is a marker of pediatric hypertension and predicts development of cardiovascular events. Electrocardiography (ECG) screening is used in pediatrics to detect LVH thanks to major accessibility, reproducibility and easy to use compared to transthoracic echocardiography (TTE), that remains the standard technique. Several diseases were previously investigated, but no data exists regarding our study population. The aim of our study was to evaluate the relationship between electrocardiographic and echocardiographic criteria of LVH in normotensive African children.

**Methods.** We studied 313 children (mean age 7,8 ± 3 yo), in north-Madagascar. They underwent ECG and TTE. Sokolow-Lyon index was calculated to identify ECG-LVH (>35 mm). Left ventricle mass (LVM) with TTE was calculated and indexed by height$^{2.7}$ (LVMI$^{2.7}$) and weight (LVMI$^w$). We report the prevalence of TTE-LVH using three methods: (1) calculating percentiles age- and sex- specific with values >95th percentile identifying LVH; (2) LVMI$^{2.7}$ >51 g/m$^{2.7}$; (3) LVMI$^w$ >3.4 g/weight.

**Results.** 40 (13%) children showed LVMI values >95th percentile, 24 children (8%) an LVMI$^{2.7}$ >51 g/m$^{2.7}$ while 19 children (6%) an LVMI$^w$ >3.4 g/kg. LVH-ECG by Sokolow-Lyon index was present in five, three and three children respectively, with poor values of sensitivity (ranging from 13 to 16%), positive predictive value (from 11 to 18%) and high values of specificity (up to 92%). The effects of anthropometrics parameters on Sokolow-Lyon were analyzed and showed poor correlation.

**Conclusion.** ECG is a poor screening test for detecting LVH in children. In clinical practice, TTE remains the only tool to be used to exclude LVH.

## INTRODUCTION

Left ventricular hypertrophy (LVH) in adults has received much attention since its detection is correlated to long-term clinical outcome, predicting cardiovascular events as myocardial infarction, stroke and death (*Devereux et al., 2001*; *Koren et al., 1991*; *Brown, Giles & Croft, 2000*; *Levy et al., 1990*). LVH results from adaptation of the heart to increased hemodynamic burden, therefore early diagnosis is important, especially in children. In the pediatric population, LVH can be used as a marker to identify hypertensive children and predict development of future cardiovascular events (*Hanevold et al., 2004*). Electrocardiographic (ECG) screening is widely used in pediatrics to detect and diagnose LVH and is considered a possible screening tool for hypertrophic cardiomyopathy (*Gersh et al., 2011*), which is responsible for almost half of sudden cardiac death cases in developed countries (*Maron et al., 1995*). Transthoracic echocardiography (TTE) is generally considered as the standard technique to diagnose LVH, but ECG seems to be more attractive as a screening tool, especially in developing countries with less resources available, thanks to its lower costs, major accessibility and good reproducibility. ECG is also easily readable by non-specialist users compared to TTE. The validity of ECG criteria for diagnosing LVH has been previously studied in several diseases, such as pediatric hypertension (*Ramaswamy et al., 2009*), rheumatic heart disease (*Sastroasmoro, Madiyono & Oesman, 1991*), hypertrophic cardiomyopathy (*Panza & Maron, 1989*), HIV infection (*Rivenes et al., 2003*), aortic stenosis and ventricular septal defects (*Fogel, Lieb & Seliem, 1995*). This correlation remains to be determined in the population of this study, which is composed by African normotensive children population.

The aim of our study was to evaluate the role of ECG as screening tool of LVH through the relationship between electrocardiographic and echocardiographic criteria in a normotensive African children population.

## MATERIAL AND METHODS

We performed a clinical, electrocardiographic and echocardiographic evaluation of 313 consecutive African children (ranging from four to 16 years old) describing the correlation between electrocardiographic and echocardiographic criteria for LVH. This study was conducted during a medical workcamp, coordinated by the Cardiovascular Department of Campus Bio-Medico of Rome, with the participation of cardiologists, pediatrics, pediatric cardiothoracic surgeons, surgeons and medical students. We had our main base at the "Clinique Médico-Surgicale St. Damien" of Ambanja, and we have investigated the region of Antsiranana, in the north of Madagascar, visiting the catholic schools of Sekoly Venance Manifatra "SE.VE.MA" and Foyer Mangafaly in Ambanja and the college Sainte Therese de l'Enfant Jesus in Maromandia, during the month of October 2015. In primary and high schools, we randomly selected a class from each group of age to enroll individuals from four to 16 years. At the "Clinique St. Damien" we enrolled children who came for routine

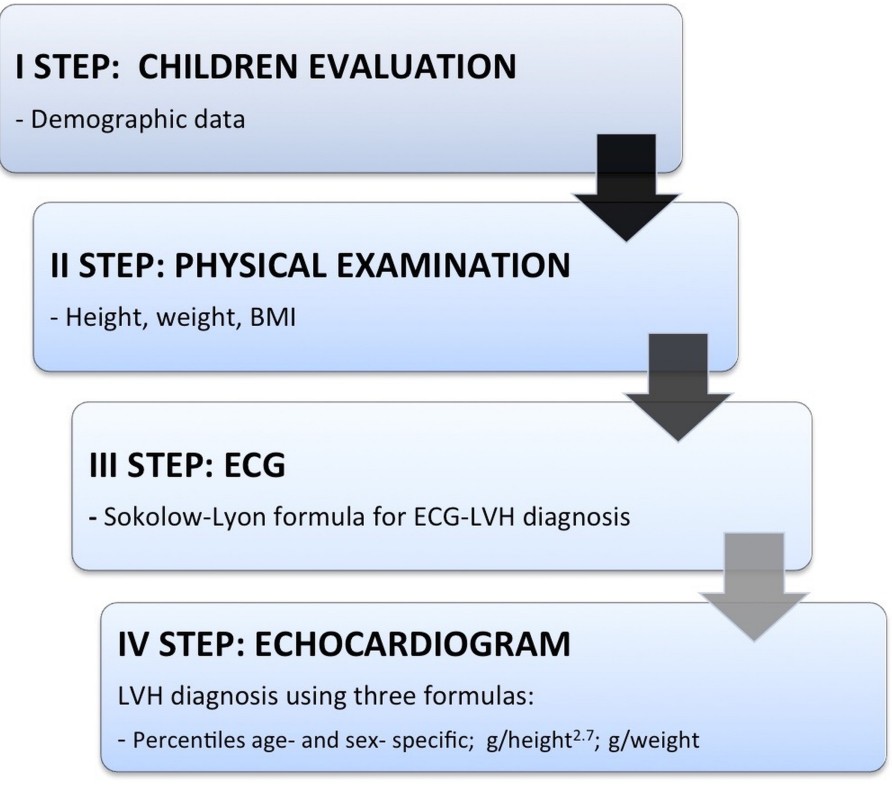

**Figure 1** Flow-chart of children's evaluation.

visits or screening, or in-patients. All individuals underwent four steps (see Fig. 1). The first step consisted in registration into our database with name, surname (when available) and date-of-birth. The second step consisted in annotation of weight, height, body mass index (BMI) calculation, blood pressure, heart rate and cardiac auscultation. Auscultation was made by a pediatrician using a standard approach. The clinical questionnaire was perfomed by a visiting medical student aided and—when necessary—by a local interpreter who has been educated in advance. Teachers or their coworkers helped the younger children understanding and answering the questions. The third step consisted in a 12-leads ECG execution and the fourth step in consisted of a complete transthoracic echocardiogram for all the study-included subjects. Medical students—supervised by cardiologists—performed ECG according to international guidelines, by using a portable electro-cardiographer General Electric, Marquette MAC 5000 (GE, Milwaukee, WI, USA) at a sampling rate of 150 Hz, at standard paper speed (25 mm/sec) and voltages (10 mm/mV). ECG voltages, representing left ventricular forces, were calculated by hand for each ECG. Non-voltage based anomalies, including T waves abnormalities were noted. Sokolow-Lyon index (SV1+RV5/RV6, mm) was calculated to identify children with electrocardiographic diagnosis of LVH (>35 mm). To calculate the left ventricular voltages, each patient's BMI was indexed to the age-based population-derived 50% normative data for age, developing the formula: Indexed ECG voltage = [ECG voltage × ($BMI_{patient}$/$BMI_{50\%}$)] (*Czosek et al., 2014*). The QTc interval was calculated with Bazett formula as follows: $QT/\sqrt{RR}$. The

upper normal limit was defined as 450 ms for boys and 460 ms for girls (*Pearl, 1996*). A team of cardiologists, with a portable echocardiograph (Esaote Mylab Five, Genoa, Italy) equipped with a S5-1 transducer probe, performed TTE. All the exams were stored on appropriate supports. Subjects were studied in the left lateral recumbent position and all standard echocardiographic views were acquired. The LV inner dimensions were measured at end-diastole and end-systole using M-Mode echocardiography in the parasternal long axis view. End-diastole was defined as the frame following mitral valve closure; while end-systole as the frame following mitral valve opening. Left ventricular mass (LVM) was calculated based on Devereux's formula (*Devereux et al., 1986*) and indexed (*National High Blood Pressure Education ProgramWorking Group on High Blood Pressure in Children and Adolescents, 2004*) by height$^{2.7}$ (LVMI$^{2.7}$) or weight (LVMI$^w$). In particular, LVM results from the formula: $LVM = 1/4\ 0.8 \times (1.04\ [LVIDd + PWTd + SWTd)^3 - (LVIDd)^3]) + 0.6$ g, where LVIDd is the left ventricular internal dimension at the end diastole, PWTd is the posterior wall thickness at the end diastole, and SWTd is the septal wall thickness at the end diastole (*Lang et al., 2015*). We report the prevalence of TTE-LVH by using three different methods (*De Simone et al., 1992*; *Daniels et al., 1995*; *De Simone et al., 1995*; *Rijnbeek et al., 2008*; *Overbeek et al., 2006*): (1) calculating the population percentiles age- and sex-specific with values above the 95th percentile identifying children with LVH; (2) LVMI value > 51 g/height$^{2.7}$ (*Hanevold et al., 2004*); (3) LVMI > 3.4 g/weight. Because of widespread illiteracy in the population, written informed consent was difficult to obtain. Indeed, a verbal informed consent—with a teacher's help to translate the language—was obtained from the children's parents, who gave study approval to the school's teachers and our research group. The study was reviewed and approved before it began by the ethics committees of the University Campus Bio-Medico of Rome (approval number 21.15 TS) and the project started in collaboration with the doctors of the hospital "Clinique Médico-Surgicale St. Damien" of Ambanja that approved the study and approved the submission to the ethics committees of the University Campus Bio-Medico of Rome (since "Clinique Médico-Surgicale St. Damien" of Ambanja" lacked an ethics committee).

## Statistical analysis

Categorical variables are expressed as frequencies and percentages in parentheses, and are compared by using Fisher's exact test or Chi-square test, as appropriate. Normality criteria were checked and met for any continuous variable, which is presented as mean and standard deviation and compared using Student $t$-test for independent data. Correlations between continuous variables were calculated using Pearson's test. Considering the correlation between Sokolow-Lyon Indexed and LVM as the primary end-point, we expected a correlation coefficient of 0.20 with an alpha error 0.05 and 90% power; this led us to calculate a sample size of 258 patients. The recruitment target was increased of 20% due to unexpected variability and final sample size was of 313 children. The sensitivity, specificity, positive predictive value (PPV) and negative predicting value (NPV) were calculated using 2×2 contingency tables. A $P$ value less than 0.05 was considered statistically significant. Statistical analysis was performed with STATA Statistics for Windows (SE, version 13).
## RESULTS

A total of 313 children of African race were studied, with a slight prevalence of female sex (53%). Mean age was 7,8 $\pm$ 3 years, ranging from four to 16 years. Clinical, electrocardiographic and echocardiographic features of population study are listed in Table 1. All children had normal arterial blood pressure. In 36 children (12%), a cardiac murmur was detected at physical examination.

At ECG evaluation, in 19% of all children sinus rhythm was found with physiological sinus arrhythmia and sinus tachycardia (mean value > 100 beats/minute) typical of the investigated range of age. Seven (2%) children with short PR segment (<120 ms) were identified. None of them had evident signs of pre-excitation or history of cardiac arrest. Abnormalities of T wave (prevalently flat T wave) were identified in 67 children (21,4%). There was no correlation between clinical evaluation and evidence of ECG abnormalities: only 14 children had both a cardiac murmur and an ECG abnormality with a 21% sensitivity of clinical evaluation to predict ECG abnormalities, a 39% PPV and a 91% specificity. The QT interval was in the normal range in all children under study. 28 children (9%) presented a Sokolow-Lyon index >35 mm, having an ECG diagnosis of LVH. I degree AV block was present in only 2 children, so extra beats. Positional Q waves were identified in 12 children while mild IV conduction delay (between 100 and 110 mesc) were present in 33 children. No bundle brunch blocks were identified.

At TTE evaluation, a mild mitral regurgitation (MR) was identified in 23 (7%) children, while three children showed moderate MR. No severe MR were identified. Five cases of mild aortic regurgitation (AR) were diagnosed with only one young boy having moderate AR. Only two children showed mitral valve prolapse of anterior mitral leaflet with mild regurgitation. Five congenital heart defects (1.6% of children) were diagnosed: two inter-atrial defects, one child with an inter-ventricular defect, one child with bicuspid aortic valve and one young girl with *cor triatriatum sinister*. Only four children with cardiac murmur presented also an echocardiographic evidence of valve regurgitation.

Following the work of *Khoury et al. (2009)*, which gave age-specific reference values for children's LVMI, the 313 children in our study were divided according to age, sex and LVM as follows: 86 children (27%) were in the $\leq$10th percentile, 32 (10%) were in the 25th, 57 (18%) in the 50th; 55 (18%) in the 75th, 30 (10%) in the 90th, 13 children (4%) in the 95th and 40 (13%) with values above 95th percentile.

The mean values of LVMI g/m$^{2.7}$ were 31,9 $\pm$ 11,8 g/m$^{2.7}$ in the overall population, with only a slight increase in children with an ECG-LVH diagnosis (32,6 $\pm$ 45,9 g/m$^{2.7}$), while the mean values for LVMI in g/kg were 2,3 $\pm$ 3 g/kg for the overall population and 2,4 $\pm$ 3,4 g/kg for the children with an ECG-LVH diagnosis. 24 children (8%) had an LVMI$^{2.7}$ > 51 g/m$^{2.7}$ while 19 children (6%) showed an LVMI$^w$ (>3.4 g/kg). Sensitivity, specificity, PPV and NPV of three investigated methods to diagnose LVH are listed in Table 2.

The capability of Sokolow-Lyon index to identify children with TTE-LVH appears to be very poor. Only five children with ECG-LVH showed also an echocardiographic diagnosis of LVH (>95° percentile); only three children were identified with other two methods. The distribution of children having an ECG-LVH according to LMVI percentiles is showed

**Table 1   Characteristics of study population.**

| | |
|---|---|
| No. of children | 313 |
| Male, *n* (%) | 146 (47) |
| Age, years | 7,8 ± 3 |
| Height, cm | 120 ± 20 |
| Weight, kg | 23,2 ± 8,2 |
| BMI, kg/m$^2$ | 15,5 ± 1,9 |
| BSA | 0,6 ± 0,3 |
| SP, mmHg | 113 ± 8 |
| DP, mmHg | 70 ± 7 |
| Heart murmur, *n* (%) | 36 (12) |
| *ECG* | |
| Sinus arrhythmia, *n* (%) | 59 (19) |
| HR, beats | 103 ± 18 |
| PR segment, ms | 140 ± 25 |
| Short PR segment, *n* (%) | 7 (2) |
| QRS complex, ms | 85 ± 13 |
| Axis, degrees | 56 ± 31 |
| Axial deviation, *n* (%) | 35 (11) |
| T wave abnormalities, *n* (%) | 67 (21) |
| LVH, *n* (%) | 28 (9) |
| Sokolow-Lyon, mm | 23,9 ± 9,2 |
| Sokolow-Lyon Indexed, mm | 24,1 ± 9,8 |
| I degree AV block | 2 (1) |
| IV conduction delay, *n* (%) | 33 (11) |
| Early repolarization, *n* (%) | 8 (3) |
| Ventricular extrasystoles, *n* (%) | 1 (0,3) |
| Supra ventricular extrasystoles, *n* (%) | 1 (0,3) |
| QT interval, msec | 364 ± 41 |
| QTc, msec | 411 ± 48 |
| *Echocardiography* | |
| LVEDD, mm | 35 ± 4 |
| IVS, mm | 6,3 ± 1,3 |
| PW, mm | 5,3 ± 0,9 |
| LVM, g | 50,5 ± 16 |

**Notes.**

The numbers are expressed as numerical values (%) or mean ± standard deviation.

Abbreviations: AV, atrio-ventricular; BMI, body mass index; BSA, body surface area; DP, diastolic pressure; HR, heart rate; IV, intra-ventricular; IVS, inter-ventricular septum; LVEDD, left ventricular end-diastolic diameter; LVH, left ventricular hypertrophy; LVM, left ventricular mass; PW, posterior wall; SP, systolic pressure.

in Table 3. Values of Sokolow-Lyon Index did not change according to LVMI percentiles (Fig. 2).

Effects of anthropometrics parameters, including body surface area (BSA), BMI, height, weight, and LVM (also indexed) on Sokolow-Lyon formula were analyzed (Fig. 3) and showed poor correlation. In the same way, when ECG voltages were indexed to patient's

**Table 2  Left ventricular mass and results of different methods of indexation according to ECG-LVH.**

| | Overall ($n = 313$) | ECG-LVH ($n = 28$) | Sensitivity | Specificity | PPV | NPV |
|---|---|---|---|---|---|---|
| LVMI >95° percentile | 40 | 5 | 13% | 92% | 18% | 88% |
| LVMI$^{2.7}$ (>51 g/m$^{2.7}$) | $31,9 \pm 11,8$ (24) | $32,6 \pm 45,9$ (3) | 13% | 91% | 11% | 93% |
| LVMI$^w$ (>3.4 g/kg ) | $2,3 \pm 3$ (19) | $2,4 \pm 3,4$ (3) | 16% | 91% | 11% | 94% |

Notes.
  The numbers are expressed as numerical values or mean ± standard deviation.
  Abbreviations:  LVH, left ventricular hypertrophy;  LVMI, left ventricular mass indexed;  NPV,  negative predicting value;  PPV,  positive predicting value.

**Table 3  Distribution of children with LVH diagnosed with ECG among LVMI percentiles.**

| | LVMI percentiles | | | | | | |
|---|---|---|---|---|---|---|---|
| | ≤10° | 11°–25° | 26°–50° | 51°–75° | 76°–90° | 91°–95° | >95° |
| ECG-LVH, $n$ (%) | 8/86 (9,3) | 4/32 (12,5) | 3/57 (5,2) | 4/55 (7,2) | 2/30 (6,6) | 2/13 (15,3) | 5/40 (12,4) |

Notes.
  Abbreviations:  LVM, left ventricular mass;  LVMI, left ventricular mass indexed.

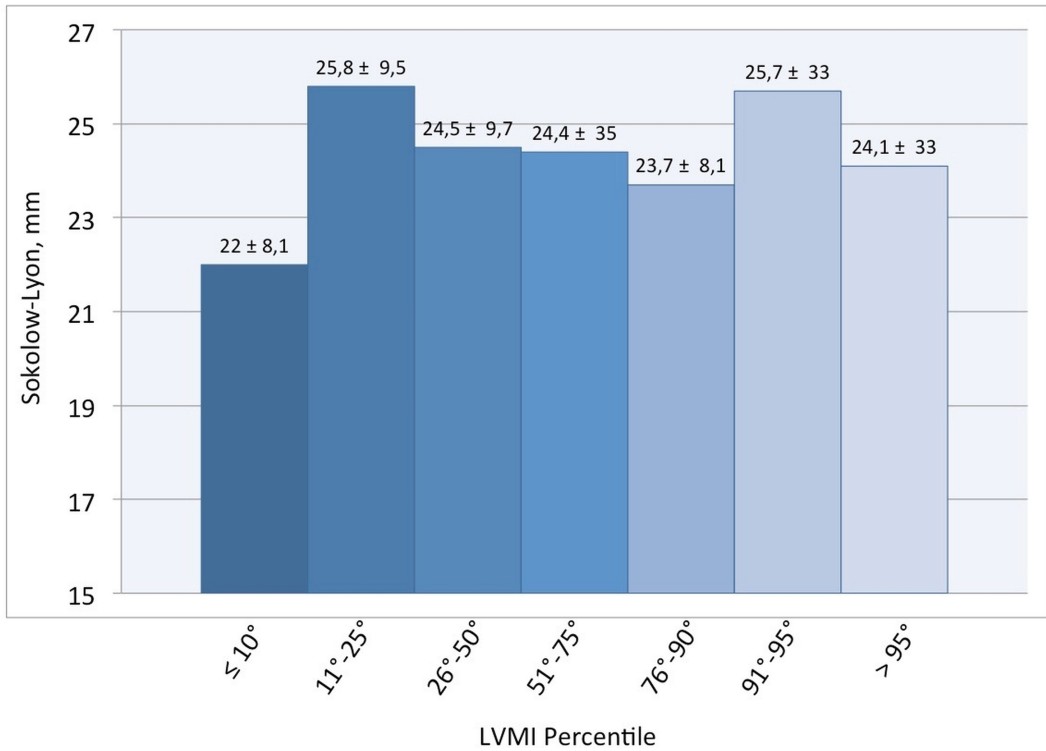

**Figure 2  Mean values of Sokolow-Lyon according to LVMI percentiles.**

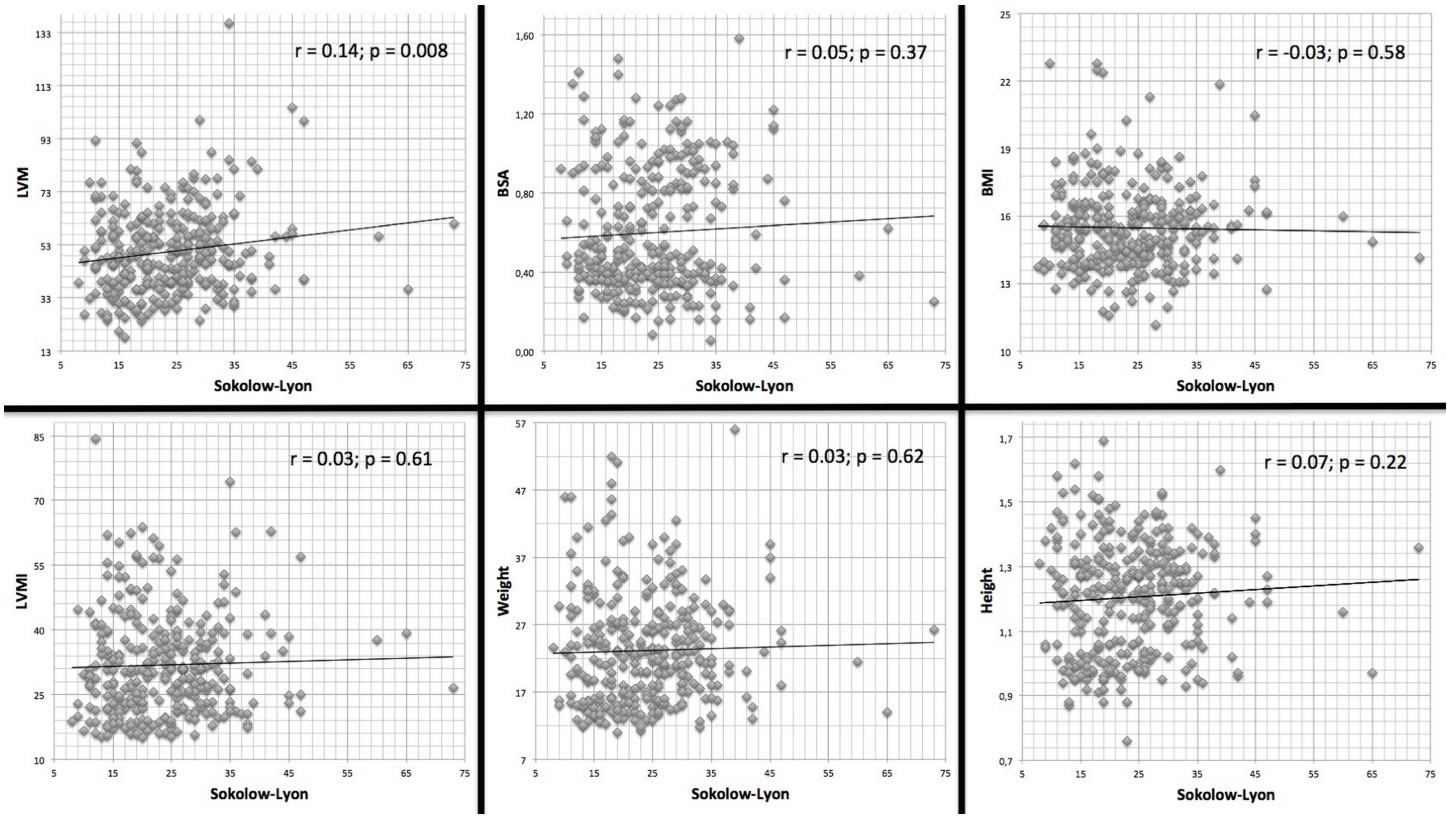

**Figure 3   Correlation between ECG Sokolow-Lyon formula with echocardiographic LV mass indices and anthropometric parameters.**

BMI (Sokolow-Lyon Indexed), direct correlation with LVM and LVMI values (Fig. 4) were not statistically significant.

Physical parameters, such as weight, height, BMI and BSA had no independent effect in the value of ECG parameters and the accuracy of the detection of LVH.

Authors acknowledge the fact that the reference value of Sokolow-Lyon > 35 mm to define LVH is used in adults. We were not able to find a better cut-off in our young population. Considering $LVMI^{2.7} > 51$ g/m$^{2.7}$ to define echocardiographic LVH, the area under the ROC curve (Fig. 5) for the electrocardiographic Sokolow-Lyon Indexed was 0.496 (95% CI [0.374–0.618], $P$ value = 0.949). Therefore, a precise cutoff in our population could not be calculated accurately.

## DISCUSSION

This study has examined the correlation and the accuracy of ECG criteria to detect TTE-LVH in a not hypertensive Malagasy children population.

Cardiac magnetic resonance imaging and 3-dimensional echocardiography have proved to be very accurate tools to diagnose and quantify LVH, but they are very expensive, they require operator expertise and their use is unachievable in less developed countries where, most of the time, it is difficult to submit population to an echocardiogram screening.

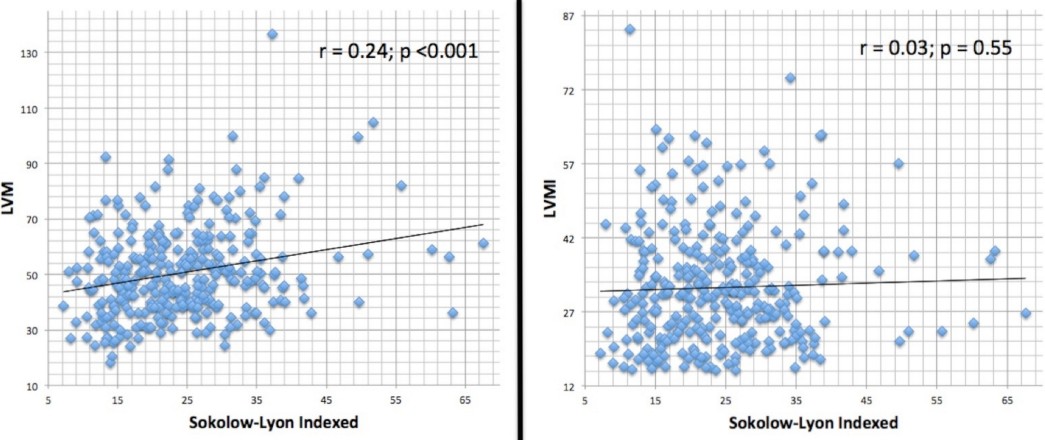

**Figure 4** Correlation between ECG Sokolow-Lyon indexed to BMI and echocardiographic LV mass indices.

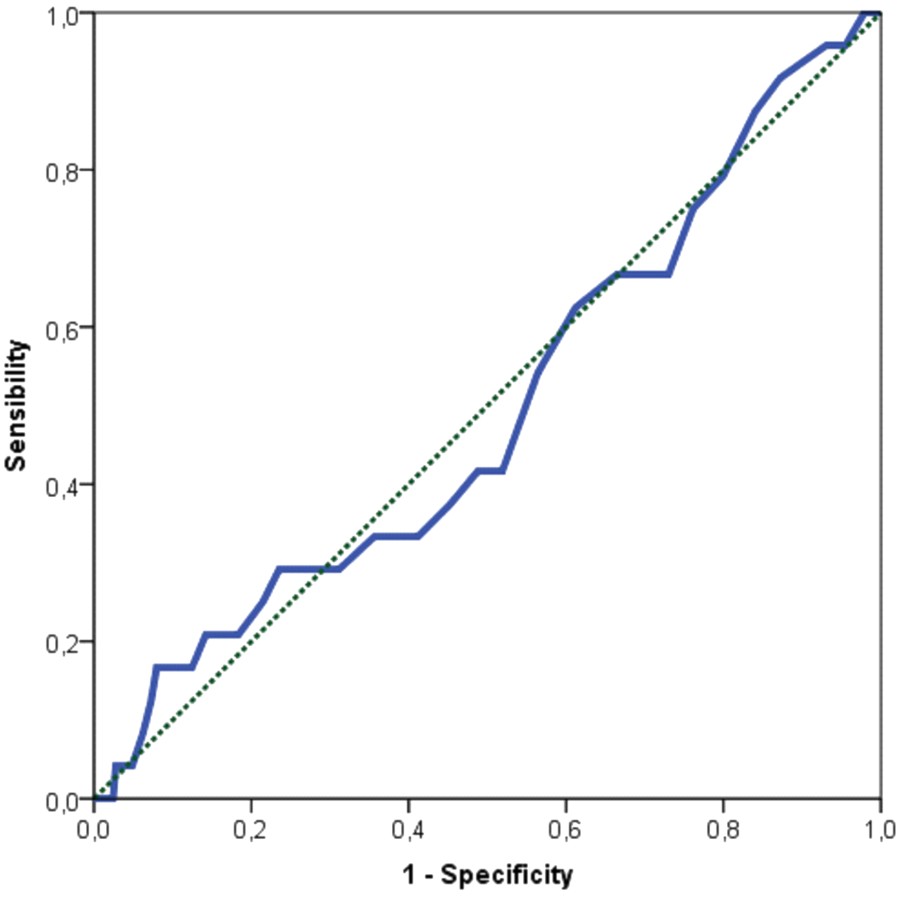

**Figure 5** ROC curve between LVMI g/m$^{2.7}$ and LVH evaluated through Sokolow-Lyon indexed.

The ECG tool has been used since a long time to investigate heart anomalies. To date, there are no clearly established benefits of a widespread, universal ECG screening in the young population (*Sharma et al., 2013*; *Fuller, 2000*). It is inexpensive compared to TTE or other imaging modalities, requires less specialist skills and takes considerably less time.

In contrast, measuring LVH by TTE M-mode technique is relatively easy and quite available, and actually the first method to be validated and currently the standard clinical diagnostic method for detecting LVH (*Alfakih et al., 2006*).

Our study used different methods for the calculation of LVM.

In a population of not hypertensive children, ECG Sokolow-Lyon criteria analysis for the detection of TTE-LVH suggests that ECG is a poor screening method for LVH.

More than 30 different ECG criteria exists for the detection of LVH on standard 12 leads ECG. In a review (*Pewsner et al., 2007*) of 21 studies including more than 5,000 hypertensive patients, using six different ECG criteria, the authors concluded that ECG criteria cannot be used to exclude LVH in adult hypertensive patients. However, many of the criteria used in adult subjects cannot be applied to a children population of different age, gender and body surface area.

Previously several studies have examined a variety of ECG parameters in children, but our type of population was not investigated before.

The discordance between ECG criteria and TTE-LVH has been described in several pediatric disease states, such as rheumatic heart disease (*Sastroasmoro, Madiyono & Oesman, 1991*), myocarditis (*Oda, Hamamoto & Morinaga, 1982*) and hypertrophic cardiomyopathy (*Panza & Maron, 1989*; *Louie & Maron, 1986*; *Maron et al., 1983*).

In children with rheumatic heart disease (*Sastroasmoro, Madiyono & Oesman, 1991*), sensitivity and specificity of ECG were 68% and 76%, respectively. Sensitivity in hypertrophic cardiomyopathy reached 76% (*Louie & Maron, 1986*).

*Rivenes et al. (2003)* investigated children with human immunodeficiency virus (HIV) infection; prevalence of ECG-LVH was 7.4% with sensitivity less than 20% and 90% specificity but used ECG-LVH criteria by Davignon (*Davignon, Rautaharju & Boisselle 1979*), which are not gender specific.

*Rijnbeek et al. (2008)* reported a study of 832 unselected pediatric hospital population using wide types of parameters for ECG-LVH detection, reporting a less than 25% sensitivity.

In diseases with pressure or volume ventricular overload as aortic valve stenosis or ventricular septal defect, *Fogel, Lieb & Seliem (1995)* found that, regardless of age, Sokolow-Lyon criteria were statistically higher compared to normal children, with the highest sensitivity in aortic stenosis patients (67%).

The study by *Morganroth et al. (1975)* found excessive values of sensitivity of ECG criteria with false-positive diagnosis in an adolescent cohort with no TTE-LVH, but LVM was not indexed to BSA.

An interesting data that emerged from our study was the high prevalence of children showing LVH, revealed both from the value of $LVMI^{2.7}$, adjusted by age and sex (24 children, 8%), from LVMI > 95th percentile (40 children, 13%) and $LVMI^w$ (19 children, 6%). In fact, the prevalence of LVH in pediatric hypertensive population has been reported

to vary from 8%–41% depending on the criteria used for determining hypertension and LVMI (*Brady et al., 2008*; *Daniels et al., 1998*; *Daniels, Meyer & Loggie, 1990*; *Sorof et al., 2002*; *Laird & Fixler, 1981*; *Niederle et al., 1982*).

ECG is an easily obtainable, low cost, rapid test but with several limitations that do not allow to substitute a TTE evaluation, even in developing countries, where less resources are available.

Nowadays, ECG screening is used in many preparticipation sports screening programs to detect cardiac abnormalities. ECG is a poor screening method for LVH, with very low values of sensitivity in general population, as demonstrated by our study in normotensive children. Other studied showed similar values for hypertensive children and, at best, modest values of specificity and sensitivity where reached, in diseases affecting directly left ventricular mass or pressure-loading conditions like aortic stenosis or hypertrophic cardiomyopathy, but non-obtaining optimal value to consider ECG as a valid screening method. Echocardiography remains the best clinical tool for LVH screening in pediatric population.

## CONCLUSION

In a normotensive African population, ECG is a poor screening test for the detection of LVH in children. In clinical practice, TTE remains the only tool to be used to exclude LVH.

### Funding
The authors received no funding for this work.

### Competing Interests
The authors declare there are no competing interests.

### Author Contributions
- Giuseppe Di Gioia conceived and designed the experiments, analyzed the data, wrote the paper, prepared figures and/or tables, reviewed drafts of the paper.
- Antonio Creta conceived and designed the experiments, wrote the paper, reviewed drafts of the paper.
- Cosimo Marco Campanale analyzed the data, reviewed drafts of the paper.
- Mario Fittipaldi, Fabio Quintarelli and Ranieri Bettini contributed reagents/materials/-analysis tools.
- Riccardo Giorgino, Umberto Satriano, Alessandro Cruciani, Vincenzo Antinolfi, Stefano Di Berardino and Davide Costanzo performed the experiments.
- Giuseppe Mangiameli analyzed the data.
- Marco Caricato analyzed the data, wrote the paper, prepared figures and/or tables, reviewed drafts of the paper.
- Giovanni Mottini conceived and designed the experiments, wrote the paper.
## Human Ethics

The following information was supplied relating to ethical approvals (i.e., approving body and any reference numbers):

Campus Bio Medico University of Rome Ethics Committee.

Approval number: 21.15 TS

Date of approval: 27/09/2015.

## Data Availability

The raw data has been supplied as a Supplemental Information.

## Supplemental Information

Supplemental information for this article can be found online at http://dx.doi.org/10.7717/peerj.2439#supplemental-information.

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
