# Peer review of "ECG is an inefficient screening-tool for left ventricular hypertrophy in normotensive African children population"

_PeerJ, doi:10.7717/peerj.2439_

## Round 0.1 · original submission · Minor Revisions

· Academic Editor

Minor Revisions

The authors in the study entitled "ECG is an inefficient screening-tool for left ventricular hypertrophy in normotensive African children population" aimed to study and evaluate the relationship between electrocardiographic and echocardiographic criteria of LVH in normotensive African children.

Based on the reviews performed by the reviewers I suggest that you improve the English language and add a flow chart which summarizes the steps followed in this study. The authors have to provide information regarding the lack of local Ethical Committee approval. Regarding other points, the authors have to improve the manuscript, following where possible, the suggestions of the reviewers.

Reviewer 1 ·

Basic reporting

Di Gioia et al., in this article try to evaluate the role of ECG as screening tool of LVH through the relationship between electrocardiographic and echocardiographic criteria in a normotensive African children population.

Comments:

The paper appear in my opinion well discussed and designed.
Each section appear clear. The authors well describe their data and the methodology used to conduct the study.
The results obtained as well as the tables and figures produced are well discussed.

Experimental design

The design of the study was well structured and discussed.

Validity of the findings

I found interesting their results.
I think the study it is very interesting, I think it is really important try to evaluate the relationship between electrocardiographic and echocardiographic criteria of LVH in normotensive African children in order to follow new strategy planning.

Reviewer 2 ·

Basic reporting

No Comments

Experimental design

The experimental design is good. It is described well, in the right way.

Validity of the findings

The findings are good even if the study population. This can be considered as a pilot study.

Additional comments

Di Gioia et al., performed a study to evaluate the relationship between electrocardiographic and echocardiographic criteria of 48 LVH in normotensive African children.
The study is well performed even if the study population is limited.
Comments:
- Line 157. .... were identified in one-fifth of the population.... I suggest to use always the same way to express the data. Before the authors used the percentage, I think they should tom convert this in percentage.
- I suggest to revise the English language.

·

Basic reporting

The manuscript of Di Gioia and colleagues has been written clearly and in unambiguous way. For what the language is concerned, in my opinion it is professional and correct, but please note that I am not English native speaker. The introductive background and state of the art clearly expose the problem and introduce it to the context. All of the figures and tables are useful, relevant and well described.

Experimental design

The research fully fits with the aims of the journal and fills the knowledge gap on the use of the ECG to detect LVH in children. Applied methods are well described, more than enough to be replicate. Results are well exposed and discussed in a concise and well explained discussion chapter.

Validity of the findings

The manuscript "ECG is an inefficient screening-tool for left ventricular hypertrophy in normotensive African children population" is an useful and interesting research on the importance of do not limit tests for detecting LVH to the ECG in children. Data are robust and all of the performed analyses are relevant and statistically significant.
In conclusion, in my opinion the research of Di Gioia and colleagues is strongly recommended for the publication in PeerJ as is.

Additional comments

No Comments

·

Basic reporting

It is a very interesting paper because majority of physician in developing countries used to diagnose LVH only by ECG criteria. The paper is well written, but requires some methodological improvements.

Author should explain why the study was not submitted in Local Ethical Comitee.

In case of diagnosis of LVH in children, author should mention what is done to these children.

Experimental design

Material and method:
a. author should writte about sample calculation as children were selected randomly.
b. author should show a flow chart which summarizes the steps followed in this study, and indicate clearly what is the reference technique among the criteria for evaluation of left ventricular hypertrophy and what is the technique evaluated.
c. Statistical analysis: author should correct positive predictive values and negative predictive values (table 2) taking into account techniques reference and technique to evaluate, and taking into account the prevalence of disease in the population (Bayes theorem).
d. The reference value for 35 in Sokolow-Lyon index is used primarily in adults. It would be better to redefine a threshold (by using a curve ROC curve for example) to study whether choosing new threshold will improve the sensitivity and specificity of the ECG to diagnose LVH in children. Author then will be able to suggest (on not) a new threshold to use Solokow-Lyon index in children.

Validity of the findings

Results
a. Table 2: the values shown in this table are not uniform: in fact, the values on the second line express the number of individuals with LVMI > 95 percentile, while on the 3rd and 4th line (LVMI2.7 and LVMIw) the main number expresses the mean and the standard deviation for the sample as a whole, while the number in brackets expresses the number of individuals above the threshold. Author should describe in the text the range, means or modal and the standard deviations of each values, and only in the table the number of individuals diagnosed being LVH, with their confidence interval.
b. Author should correct absolutely the PPV and NPV
3. Discussion
The prevalence of LVH in children varies depending on the criteria used. Author should develop what are the impacts in terms of decision in therapeutic support. Author should elaborated in the text indication of the use of these different criteria.

Minor revision:
Author should review (MR & RA) abbreviations: the meaning is not mentioned in the text.
Author should correct negative predictive value (NPV) but no NPP.

Additional comments

The paper needs only some additionnal statistics analysis and some corrections of forms to be published.

---

## Round 0.2 · Minor Revisions

· Academic Editor

Minor Revisions

I have only one comment :

I suggest to the authors to insert a sentence in the manuscript, in which it is explained that "the study was reviewed and approved before it began by ethics committees of University Campus Bio-Medico of Rome and the project started in collaboration with the doctors of hospital “Clinique Médico-Surgicale St. Damien” of Ambanja that approved the study and approved the submission to the ethics committees of University Campus Bio-Medico of Rome " (since the lack of an Ethical committee in the hospital “Clinique Médico-Surgicale St. Damien” of Ambanja").

---

## Round 0.3 · accepted · Accept

· Academic Editor

Accept

The authors performed all the required comments and the manuscript entitled "ECG is an inefficient screening-tool for left ventricular hypertrophy in normotensive African children population" is now ready for publication.

Reviewer 2 ·

Basic reporting

No Comments

Experimental design

No Comments

Validity of the findings

No Comments

Additional comments

The authors adequately addressed the concern raised for the original submission.